# Antiperthite and Mesoperthite Exsolution Textures in the Zhengjiapo BIF, Changyi Metallogenic Belt, North China Craton: Evidence of UHT Metamorphic Overprint

Yan-Rong Chen [1], Xu-Ping Li [1,*], Zeng-Sheng Li [2], Hans-Peter Schertl [3] and Fan-Mei Kong [1]

1 Shandong Provincial Key Laboratory of Depositional Mineralization and Sedimentary Minerals, Shandong University of Science and Technology, Qingdao 266590, China
2 Shandong Institute of Geological Sciences, Jinan 250013, China
3 Institute of Geology, Mineralogy and Geophysics, Faculty of Geosciences, Ruhr-University Bochum, 44780 Bochum, Germany
* Correspondence: lixuping@sdust.edu.cn

**Abstract:** Paleoproterozoic banded iron formation (BIF) iron ore of the Zhengjiapo region of the Changyi metallogenic belt, Eastern Block of North China Craton contains abundant coexisting antiperthite and mesoperthite textures. The antiperthite and mesoperthite occur in felsic domains of the Zhengjiapo BIF ore and enable derivation of peak temperature metamorphic conditions. Thermodynamic phase modeling shows that equilibrium conditions of corresponding textures, considering the related mineral assemblage of Pl + Qz + Kfs + Mag + Opx + L, are in the range of 870–940 °C and 5.0–6.8 kbar. Ternary feldspar thermometry using reintegrated compositions of antiperthite and mesoperthite in the felsic domain of the studied BIF iron ore reveals even higher peak metamorphic temperatures of 1045–1080 °C. The ultra-high temperature–low pressure conditions of Precambrian BIF have not yet been reported from the North China Craton. The documented ultra-high temperature metamorphism of the Zhengjiapo BIF iron ore in the Changyi metallogenic belt indicates that the BIF was involved in the collision-related tectonic process during Paleoproterozoic to have occurred in the Jiao-Liao-Ji orogenic belt.

**Keywords:** BIF iron ore; antiperthite and mesoperthite exsolution; UHT metamorphism; Jiao-Liao-Ji orogenic belt

## 1. Introduction

The Paleoproterozoic Jiao-Liao-Ji Belt (JLJB), located in the Eastern Block of the North China Craton (NCC) is bordered by the Bo Sea in the north and the Tan-Lu Fault in the west (Figure 1a); toward the southeast the Sulu HP-UHP orogenic belt is located. NCC is one of the oldest cratons on Earth, preserving a large number of late Archean to early Proterozoic banded iron formation (BIF) iron deposits. Most BIF iron deposits have undergone greenschist to amphibolite facies metamorphism; only a few reached granulite facies metamorphic conditions [1–5]. Some types of BIF iron deposits occur in Archean granite-greenstone belt, such as those in the Yishui complex along the western margin of the JLJB [2] and in the Taihua Group on the southern margin of the NCC [4]. Others exist in Paleoproterozoic metamorphic volcano-sedimentary sequences, such as the Changyi BIF iron deposit in Fenzishan Group of the JLJB [1] and BIFs in the Miyun Complex of the Eastern Block [5] (Figure 1a). In the following, we use the term "BIF iron ore" to distinguish it from BIF manganese ore. The Zhengjiapo BIF iron deposit is located in the Jiaobei terrane of the JLJB in the NE-SW Laizhou-Changyi iron metallogenic belt or the Changyi iron metallogenic belt (red star in Figure 1b,c). It is believed to occur in the Fenzishan Group according to previous studies [1,3,4]. Previous studies have shown that the lithostratigraphy of the Fenzishan Group ranges from a basal clastic-rich and bimodal-volcanic

sequence through a middle carbonate rich sequence to an upper pelite-rich sequence which underwent metamorphism from amphibolite facies to granulite facies conditions [1,3] and which is composed of magnetite ore, magnetite quartzite, biotite leptynite, biotite-diopside leptynite, and garnet-bearing biotite-diopside gneiss. At present, more than 20 medium and small iron ores bodies have been found in this belt. Fluid inclusion studies showed that rocks of the Changyi BIF iron deposit experienced metamorphic conditions with temperatures exceeding 636 °C [1,6].

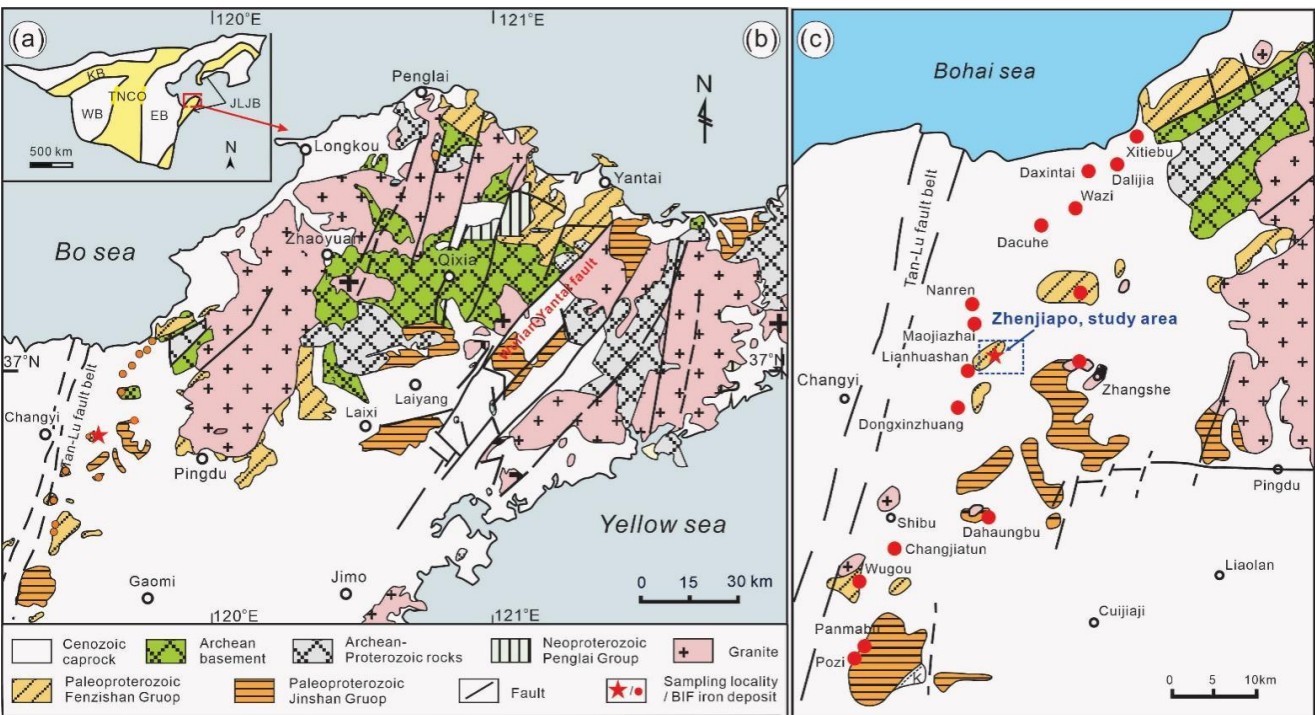

**Figure 1.** (**a**) Geological sketch map of the North China Craton (NCC) and (**b**) geological map of the Jiaobei terrane (after [3]), (**c**) sampling locality in the Zhengjiapo BIF iron mine and distribution of BIF iron ore in the Changyi metallogenic belt (modified after [1]). WB = Western Block; EB = Eastern Block; KB = Khondalite Belt; JLJB = Jiao-Liao-Ji Belt; TNCO = Trans-North China Orogen.

Ultra-high temperature (UHT) metamorphism is an important research field in metamorphic geology, which is of great significance for understanding the tectonothermal evolution of the crust [7–9]. More than dozens of UHT granulite outcrops have been identified worldwide and there are increasing reports of new discoveries or upgrades to UHT granulites [10,11]. The geological time span of UHT metamorphism can range from 3178 Ma in Mesoarchean to 35 Ma in Cenozoic and is mostly related to collision orogeny and extensional tectonic belts. The occurrence of large-scale UHT metamorphic belts from the Neoarchean to Cambrian orogenic belts implies a significant change in geodynamic regime during the Neoarchean [9,11–13].

Very few studies worldwide have shown that BIFs and/or their wall rocks experienced ultra-high temperature (UHT) metamorphism, which mostly occurred in Archean and Paleoproterozoic [8,14,15]. Archean BIF of the Voronezh Crystalline Massif which forms part of the East European Craton, contains coexisting exsolution textures of clinopyroxene and orthopyroxene in associated magnetite quartzites, having reached peak metamorphic conditions ~1000 °C/10–11 kbar under a thickened continental crust [14]. The existence of antiperthite, perthite, and mesoperthite in related metapelite from the Voronezh Crystalline Massif also indicated UHT metamorphic conditions exceeding 1000 °C [8]. The Th-U-Pb dating of monazite from these metapelitic schists revealed an age between ~2015 and 2039 Ma, which was interpreted as a regional metamorphic event that occurred contemporaneously with an intrusion of collisional S-type granitic melts about 2020 Ma ago [7]. A

further example is a meta-iron stone of the Sutherland greenstone belt, South Africa where the coexistence of clinopyroxene and pigeonite was used to document UHT metamorphic temperatures > 1000 °C; the corresponding pressure was estimated to be 10 kbar [15].

In general, BIFs of the NCC are reported to have experienced only greenschist to amphibolite facies metamorphic conditions and to be mainly distributed in greenstone belts [16]. The aim of the current study is to examine, if abundant, antiperthite and mesoperthite textures observed in felsic domains of BIF iron ores from the Zhengjiapo ore mine, Changyi iron metallogenic belt (Figure 1a,b). Related petrological studies may indicate whether the rocks have experienced similar or different metamorphic conditions, and thus more in-depth insights into possible geodynamic scenarios and implications are expected. We applied phase equilibria modeling and two-feldspar thermometers to estimate peak temperatures and to at least receive indications of the prevailing pressures.

## 2. Geological Background and Sample Description

Three Paleoproterozoic orogenic belts developed on the Archean basement of the North China Craton (NCC) [17]. The Khondalite Belt (KB) and the Jiao-Liao-Ji Belt (JLJB) occurred within the West Block (WB) and East Block (EB), respectively, while Trans-North China Orogen (TNCO) resulted in the final amalgamation of the NCC [16,17]. In other words, the NCC was welded together by these three major Paleoproterozoic collisional belts (Figure 1a). The NE-SW trending Paleoproterozoic JLJB is a collisional orogenic belt in the EB of the NCC. It underwent extensional rifting during 2.2–1.95 Ga, which led to the opening of an incipient ocean and divided the Eastern Block into the Longgang and Nangrim blocks [3,17]. During subsequent subduction and collision, these two blocks were re-assembled to form the JLJB of ~1.9 Ga [17,18].

The Jiaobei terrane is located in Shandong Peninsula and belongs to the southern part of the Jiao-Liao-Ji Belt (Figure 1a,b). It is made up of Archean basement and Paleoproterozoic metasedimentary sequences with associated granitic and mafic intrusions [3,16–19]. The Paleoproterozoic Fenzishan and Jingshan Groups directly overlie the Archean-Paleoproterozoic basement rocks with the ductile shear zone developing along the contact boundary, mainly composed of biotite leptynite, sillimanite-biotite schist, magnetite quartzite, marble, and amphibolite [3]. It is generally believed that the Fenzishan Group underwent amphibolite facies metamorphism, and its P–T condition is lower than that of the Jingshan Group [3,16,17].

The Zhengjiapo BIF iron deposit, hosted in the Fenzishan Group of the Changyi metallogenic belt, was formed in the Paleoproterozoic JLJB (Figure 1b) [1,3,17]. However, the origin of Changyi BIFs remains controversial. Combined with regional geological survey and wall rock protolith provenance studies of the BIF, Lan et al. suggested that the Changyi BIF deposition formed in a continental rift environment [1,19,20]. Wang et al. [6] argued that the BIFs in the Laizhou-Changyi or Changyi area were formed in a tectonic setting related to an island arc and were not attributed to the rift tectonic environment during the formation of Jiao-Liao Basin. In addition, based on the U-Pb age of the detrital zircon in the wall rock of the biotite plagioclase leptynite, Lan et al. [1,20] suggested that the Changyi BIF was deposited during 2240–2193 Ma and underwent amphibolite facies metamorphism at about 1864 Ma. This result is consistent with the understanding of the formation and metamorphism of Fenzishan and Jingshan Groups in JLJB as well as the formation and evolution of the entire Paleoproterozoic JLJB [3,16–18]. Nevertheless, Wang et al. [6] studied detrital zircons and obtained U-Pb ages of ~2.73 Ga and ~2.9 Ga from meta-volcanic rocks and metapelite and ~1850 Ma from amphibolite, suggesting that the source material of Changyi BIFs is completely different from that of Fenzishan Group in the JLJB. The formation age of Changyi BIFs should be in the early Neoarchean (~2.7 Ga). Therefore, the Xiaosong Formation containing Changyi BIFs should be disintegrated from the Fenzishan Group and redefined as the early Neoarchean (~2.7 Ga) sedimentary formation. Based on our study of the geochronology of the Zhengjiapo BIF and its wall rock, it is believed that there is a group of Paleoproterozoic magmatic zircon age (~2.2 Ga), which indicates that the

Changyi BIF belongs to the Fenzishan Group and is a product of the JLJB formation [20,21]. The ages of 2.7 Ga and 2.9 Ga should be derived from Archean basement rocks inside and on both sides of the JLJB.

The iron-bearing rock series in the Fenzishan Group of the Zhengjiapo iron mine are generally consisting of magnetite-biotite ± garnet leptynite, magnetite quartzite and ±garnet-biotite-amphibole-magnetite ore. The pelitic gneiss is mainly biotite-leptynite and biotite-amphibole ± garnet leptynite. BIF iron ore samples of this study were taken from the metamorphic supracrustal sequences of the Fenzishan Group that directly overlies the Archean rock series of the Zhengjiapo BIF iron mine [17,18]. The studied iron ore samples are mainly composed of magnetite and quartz associated with amphibole, biotite, plagioclase, K-feldspar, epidote, and lesser amounts of orthopyroxene, whereas pyrite, hematite, chlorite, apatite, and calcite are accessories. The distribution of minerals is heterogeneous both in the outcrop and in thin section scale, with some areas essentially rich in magnetite and others rich in quartz and feldspar (Figure 2a–i). The samples studied show crystalloblastic texture and are mainly composed of (in vol.%) magnetite (40%–50%), quartz (20%–30%), amphibole (10%–20%), biotite (5%–10%), plagioclase (5%–10%), or epidote (~5%) in areas rich in ore minerals; and of magnetite (~10%–20%), quartz (30%–40%), biotite (10%–20%), plagioclase (15%–25%), perthite (10%–15%), amphibole (10%–15%) and minor epidote (~5%), orthopyroxene (<5%) in felsic areas (Figure 2a–g). Accessory minerals are pyrite, hematite, chlorite, apatite, and calcite. In order to facilitate the description, we named the ore mineral rich area as magnetite-rich domain and the gangue mineral rich area as felsic-rich domain.

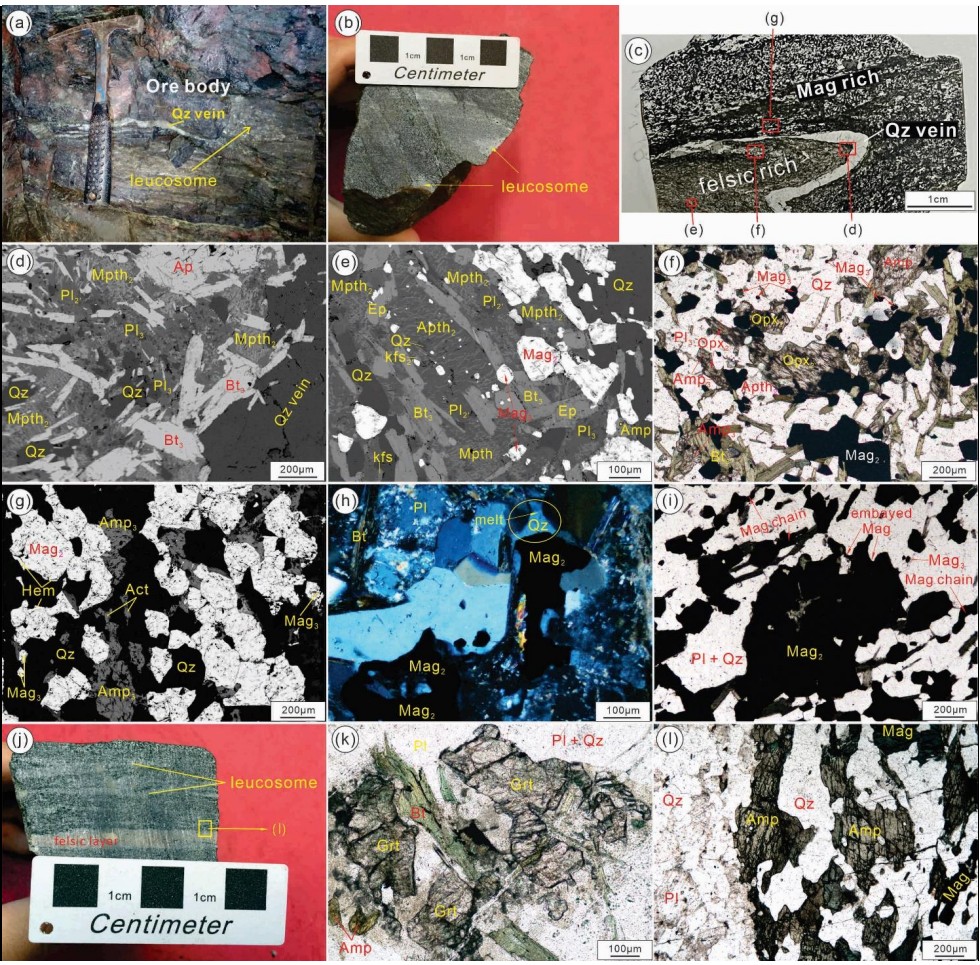

**Figure 2.** (**a**) Underground mining outcrop; (**b**) hand specimen showing leucosomes demonstrating partial melting; (**c**) thin section of a studied sample showing heterogeneous distribution of ore-mineral

rich and felsic-rich domains; (**d**,**e**) felsic-rich domain; (**f**) and (**g**) Mag associated Opx and Amp; (**h**) melt texture at triple junction of quartz and plagioclase (see [22,23]); (**i**) magnetite chain textures in the matrix [24,25] as well as embayed texture indicating the effects of high-grade metamorphism and melting (see [23,26]); (**j**–**l**) magnetite-amphibole-garnet gneiss associated with BIF iron ore; (**j**) hand specimen showing leucosome which indicates partial melting; (**k**) pelitic gneiss showing occurrence of garnet crystalloblast; (**l**) gneissic texture and embayed texture showing melting effect. Photos (**d**,**e**,**g**) are back scattered electron (BSE) images, (**f**,**i**,**k**) and (**l**) were taken under plane-polarized light; (**h**) was taken under cross-polarized light. Apth = antiperthite; Mpth = mesoperthitic; all other mineral abbreviations are taken from [27].

The rocks directly associated with BIF iron ore are magnetite-amphibole garnet gneiss with a fine-grained gneiss structure, composed of magnetite (~5%–20%), amphibole (~10%–20%), biotite (~5%–20%), plagioclase (~10%–30%, including <5% albite), K-feldspar (<10%), quartz (~10%–30%), orthopyroxene (<8%), and variable garnet (0%–15%) (Figure 2j–l). Accessory minerals are ilmenite, titanite, zircon, and apatite (Figure 2j–l).

## 3. Analytical Methods

Representative mineral compositions were determined using a JEOL JXA-8100 electron microprobe (EMP) at the Shandong Academy of Geological Sciences, Jinan, China. The EMP was operated using an accelerating voltage of 15 kV, probe current of 10 nA, and beam size of 1 μm. Calibration was conducted with the following standards: Quartz for Si, rutile for Ti, corundum for Al, chromite for Cr, hematite for Fe, spessartine for Mn, periclase for Mg, wollastonite for Ca, albite for Na, and orthoclase for K. Representative microprobe analyses of antiperthite/mesoperthite are presented in Table S1, mineral compositions of the assemblage used to calculate peak temperature conditions and compositions of additional minerals of the BIF sample are listed in Tables S1 and S2, respectively. Mineral abbreviations used in this paper are taken from [27].

## 4. Metamorphic Stage and Mineral Chemistry

Since the iron ore portion of the samples usually hosts simple mineral assemblages, such as Mag + Hem + Qz + Amp + Act, it is not possible to determine the entire P–T path. However, taking into account the intergrown felsic-rich domains, mineral assemblages at different metamorphic stages can be distinguished, which allows for the construction of a joint P–T path for the BIF iron ore). Based on petrography, mineral assemblage, and textural relationship, two metamorphic stages can be distinguished (Figure 2). In the absence of residual prograde metamorphic minerals (M1 stage), the mineral assemblage associated with antiperthite and mesoperthite is defined as the M2 metamorphic stage. Since the re-integrated ternary feldspar compositions of antiperthite and mesoperthite record even higher temperature conditions, the actual M2 mineral assemblage is taken as M2′ stage. The retrograde mineral assemblage represents a M3 metamorphic stage. Therefore, the M2 stage is characterized by the presence of antiperthite and mesoperthite, the mineral assemblage of the M2′ stage is Opx + Kfs + Pl + Qz + Mag + L; while that of M3 stage is Bt + Amp + Opx + Kfs + Pl + Qz + Mag. Detailed description is as follows:

Magnetite reveals a total content from 98.85 wt.% to 99.78 wt.% with an average value of 99.56 wt.% in the ore-rich domains and 97.79 wt.% to 99.47 wt.% with an average value of 98.75 wt.% in the felsic-rich domains (Table S2). The felsic-rich domains are dominated by silicate minerals including plagioclase, K-feldspar, mesoperthite/antiperthite, amphibole, biotite, epidote, and quartz.

Plagioclase occurs as crystalloblast in small grain sizes of 0.2–0.3 mm and in direct contact with mesoperthite and/or antiperthite, representing the near-peak metamorphic stage (M2′) (Figure 2d,e), or as matrix mineral.

K-feldspar solid solutions occur as acicular crystals along with mesoperthite and antiperthite belonging to the M2′ stage; note that it also forms a generation of fine-grained crystalloblasts or is associated with biotite belonging to the M3 stage (Figure 2e).

Mesoperthite and antiperthite are composed of K-feldspar lamellae exsolved from the matrix plagioclase with a grain size of 0.2–0.5 mm in length, formed during cooling of BIF ore body at the M2′ stage (Figure 2d,e). They are observed and surrounded by retrograded biotite which developed along filiation or which are part of the M3 stage and occur at the boundary of plagioclase and quartz (Figure 2d–f).

Biotite in the sample mainly shows tabular crystals forming a foliation in intensively deformed domains (Figure 2d,e). It may also occur in arbitrary orientation in weakly deformed regions (Figure 2f). Biotite is a retrogressive mineral occurring only at and after M3 metamorphic stage.

Orthopyroxene has only been observed at the margins of magnetite, occurring at M2′ stage (Figure 2f). Most orthopyroxenes may have been replaced by amphibole during the retrograde M3 stage, which can be seen in the retained texture (Figure 2f).

Amphibole is mainly developed around or associated with magnetite (Figure 2g) or replaces orthopyroxene at its margin (Figure 2f). Amphibole can be chemically divided into two species which are Mg-hornblende and actinolite, respectively, with both represented in retrograde minerals which occur at and after the M3 metamorphic stage.

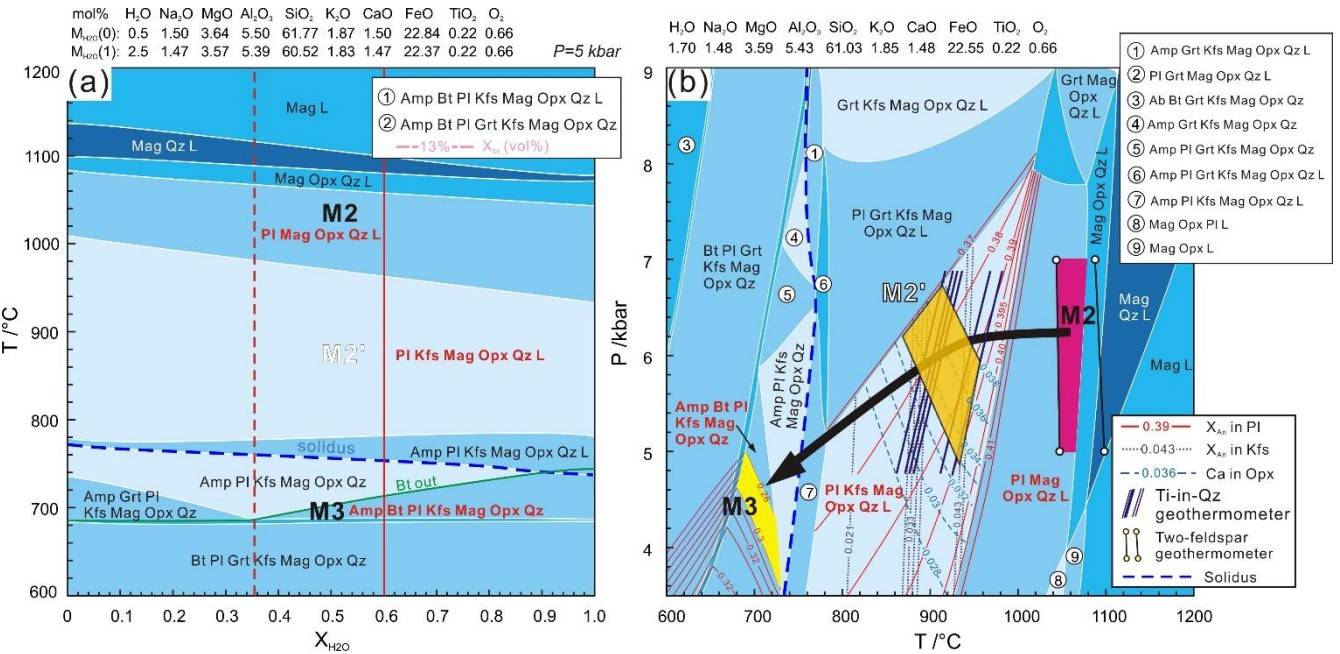

**Figure 3.** (**a**) T-$X_{H2O}$ diagram at 5 kbar, the observed related mineral assemblages are given in red and stable in between $X_{H2O} > 0.35$ and $X_{H2O} < 0.6$. Blue dashed line is the solidus. (**b**) P–T pseudosection calculated from effective bulk composition for peak temperature P–T conditions of felsic-rich domain of BIF ore sample 19CY18.

Epidote is a metamorphic mineral of about 0.1 mm in size and often overprinted by biotite, indicating its formation after the M3 stage (Figure 2e).

Quartz is present in the matrix as anhedral or elongated crystals in all stages of metamorphism. It is associated with mesoperthite and antiperthite in M2 stage, and is associated with biotite in the M3 stage (Figure 2d,e). Quartz, in particular, is vein-shaped and has undergone the whole process of prograde, peak stage, and retrograde metamorphism; it may show deformation due to collisional events (Figure 2a,c,d).

Magnetite forms subhedral-euhedral crystals, which also occurs in all stages of metamorphism. It appears as fine-grained crystals in the mesoperthite/antiperthite intergrowth textures (Figure 2e), is associated with orthopyroxene and amphibole, or appears in the matrix as crystalloblast of about 0.25–0.5 mm in M2 stage (Figure 2e–g,i). It occurs as small grains of less than 0.1 mm in the M3 stage (Figure 2e–g).

Based on textural observation and mineral relationship, three major metamorphic stages can be distinguished: The peak metamorphic M2 stage is mainly characterized by the occurrence of mesoperthite and antiperthite, which are reintegrated into the P–T conditions of plagioclase before K-feldspar became exsolved from crystal lattice. M2′ stage defined here is characterized by Pl + Kfs + Opx + Qz + Mag + L (melt), which represents near-peak P–T conditions but slightly lower than the M2 stage. The stage (M3) is comprised of Amp + Bt + Pl + Kfs + Qz + Mag; the metamorphic stage (M4) is represented by the occurrence of actinolite + epidote mineral assemblage.

The foliated biotite in the deformed regions reveals a schistose texture, which was formed most likely contemporaneously with folded quartz veins during high-grade metamorphism (Figure 2c–e). Although the foliated texture is primarily determined by biotite, likely prior to peak metamorphism, biotite can form along the mineral boundary during retrograde metamorphism, as shown in Figure 2d,e. In any case, since Fe and Mg in biotite can diffuse efficiently during metamorphism at high temperatures, the chemical composition of biotite retained is likely that in equilibrium with the assemblage M3 stage formed during retrograde metamorphism [23,26]. Prograde minerals are known to be commonly preserved inside of robust minerals, such as garnet, zircon, etc. [28,29].

Since no residual prograde minerals (M1) have been found, some plagioclase grains that contain exsolution features of K-feldspar forming mesoperthite/antiperthite, which are named as $Mpth_2/Apth_2$, represent the peak metamorphic stage M2 (Figure 2d–f). The crystalloblastic plagioclase ($Pl_2'$) (~0.2–0.5 mm) is in equilibrium with mesoperthite and antiperthite representing near-peak metamorphism stage M2′; the corresponding values of $X_{An}$ are between 0.37 and 0.43. The plagioclase ($Pl_3$) not in contact with $Mpth_2/Apth_2$ is in equilibrium with amphibole and biotite. The $X_{An}$ of this plagioclase that belongs to stage 3 (Figure 2d,e; Table S1) ranges from 0.26 to 0.30. K-feldspar, which mainly occurs as intergrowth textures with plagioclase as perthite, but several fine-grained crystals ($Kfs_2'$) associated with mesoperthite/antiperthite and magnetite occur in the matrix, exhibiting $X_{An}$ values between 0.021 and 0.043 (Figure 2e, Table S1). Crystalloblastic orthopyroxene ($Opx_2'$) is also observed in equilibrium with mesoperthite and antiperthite with $Mg^{\#}$ from 0.48 to 0.52 and Ca from 0.033 to 0.038 (a.p.f.u.) (Table S1). It is stable at near-peak temperature metamorphic conditions (M2′). Early large crystalloblastic amphibole ($Amp_3$) (0.1–0.5 mm) mostly appears along the rims of magnetite (often in direct contact with magnetite). It has a composition of Mg-hornblende, whereas late fine-grained amphibole (<0.1 mm) is actinolite (Figure 2e–g). Magnetite is stable in both stages of M2 and M3 (Figure 2e–g). Although lath-shape biotite appears to be structurally related to mesoperthite and antiperthite, it crosscuts mesoperthite and/or antiperthite (Figure 2d,e) and actually belongs to the retrograded M3 stage and is associated with $Amp_3$ (Figure 2f). It has a grain size of 0.2–1 mm with $Mg^{\#}$ from 0.50 to 0.53 (Table S2). Epidote usually grows after biotite (Figure 2e) and contains $Fe^{3+}$ between 0.72 and 0.82 (a.p.f.u.) (Table S2).

Due to the high content of magnetite in the BIF ore sample, it is relatively dry and, in general, melting textures are ambiguous. However, a careful study can still point to potential textures, such as (1) leucosome in the hand specimen (Figure 2b,j); (2) melt interstitial texture at the triple junction of quartz and plagioclase grains (Figure 2h [22,30]); (3) magnetite chain textures in the felsic matrix [24,25] as well as magnetite or amphibole embayed textures (Figure 2i,l, see [23,26]), which may indicate the influence of high-grade metamorphism and melting. Quartz veins are related to metamorphism rather than primary sedimentary bedding or formed in post-peak metamorphism under extension setting from fluids (Figure 2a,c). The veins clearly underwent compressional deformation, and may correspond to the regional collisional event in the JLJB (see discussion).

In order to figure out P–T conditions of the BIF ore, the sample 19CY18 which contains antiperthite and mesoperthite textures, was selected, and different related compositions of amphibole, orthopyroxene, biotite, plagioclase, K-feldspar, and quartz were used to derive different metamorphic stages and construct a part of a P–T path.

## 5. Estimation of Metamorphic Conditions

### 5.1. Phase Equilibrium Modeling

We applied the GeoPS software [31] and the internally consistent thermodynamic data set ds633 [32] in the NCKFMASHTO ($Na_2O$—$CaO$—$K_2O$—$FeO$—$MgO$—$Al_2O_3$—$SiO_2$—$H_2O$—$TiO_2$—$Fe_2O_3$) system for phase equilibrium modeling. The mineral-phase model of the activity–composition relationship used is taken from [32–34]. Whole rock compositions had been determined by X-ray fluorescence spectrometry (XRF) at Wuhan Sample Solution Analytical Technology Co., Ltd., Wuhan, China (Table S3). Phase equilibrium modeling calculations performed were carried out on homogeneous portions of the rock, which is a prerequisite to obtain reliable information [35].

In order to construct a P–T pseudosection, the effective bulk composition was selected to model the phase equilibrium by point counting individual minerals in the felsic domain of the magnetite ore (sample 19CY18). The estimated volume percentage of each mineral in the felsic-rich domain is obtained by combining point counting of individual minerals and the corresponding compositions were measured by electron microprobe [36]. The corresponding mineral content is: Quartz 30 vol.%, plagioclase 14.5 vol.%, biotite 13 vol.%, amphibole 12 vol.%, K-feldspar 7 vol.%, magnetite 15 vol.%, orthopyroxene 4 vol.%, and epidote 4.5 vol.%, and the corresponding effective composition in mole percentage (mol%) is: $SiO_2$ = 61.03, $Al_2O_3$ = 5.43, $TiO_2$ = 0.22, FeO = 22.55, MgO = 3.59, CaO = 1.48, NaO = 1.48, $K_2O$ = 1.85, $H_2O$ = 1.7, $O_2$ = 0.66.

A T-$X_{H2O}$ diagram is constructed to determine the appropriate $H_2O$ content at a pressure of 5 kbar on the basis of the effective bulk composition (Figure 3a). It is based on the following necessary constraints: (1) At $X_{H2O}$ > 0.35 (dashed red line), mineral assemblage of M3 is reacting out; (2) the solid red line representing an $X_{H2O}$ value of ~0.6 mol% was determined by the modeling process of the effective bulk composition.

The P–T pseudosection (Figure 3b) is established in the range of 3.5–9 kbar and 600–1200 °C. The near-peak temperature mineral assemblage (defined as M2′) in equilibrium with mesoperthite/antiperthite is Qz + Pl + Kfs + Opx + Mag + L, which resulted from the petrographic analysis as shown in Figure 2d,e. Melting textures were observed in the hand specimen and in thin sections (Figure 2b,h–i), although it is difficult to distinguish them from the monomineralic quartz veins (Figure 2c) formed from metamorphic differentiation. The modeling results of the P–T pseudosection are presented in Figure 3b. An extended temperature condition was derived (from the yellow to the pink areas in Figure 3b) and quartz, plagioclase, and magnetite are stable minerals within these P–T regions. The $X_{An}$ isopleths of plagioclase and K-feldspar are moderately to steeply inclined, vary significantly for both temperature and pressure, and can be used to constrain the P–T conditions. Whereas the $X_{An}$ isopleths in K-feldspar are nearly perpendicular to the *x*-axis (and can be essentially used to limit temperatures), the $X_{An}$ isopleths in plagioclase display a negative slope. The metamorphic conditions of this respective stage M2′ are bracketed by the isopleths of $X_{An}$ in plagioclase (0.37–0.43) and $X_{An}$ in K-feldspar (0.021–0.043, see Figure 3b).

Furthermore, Lindsley [37] designed a Ca-Mg-Fe pyroxene thermometer using experiments based on calculated phase equilibria for diopside-enstatite and hedenbergite-ferrosilite exchange reactions, which can be applied to a P–T range of 1–15 kbar/800–1200 °C. Therefore, the isopleths of Ca in orthopyroxene were shown to independently frame pressure/temperature conditions (Figure 3b); the respective values range between 0.033 and 0.038 (a.p.f.u.). Considering phase equilibrium modeling and the above data using $X_{An}$ in plagioclase and K-feldspar as well as Ca-isopleth in Opx, the P–T conditions of this M2′ stage are constrained to 870–940 °C/5.0–6.8 kbar (see framed area by green dashed line in Figure 3b).

The retrograde metamorphic stage M3 is characterized by the occurrence of amphibole ($Amp_3$), biotite ($Bt_3$), and fine-grained plagioclase ($Pl_3$) in the matrix (Figure 2c–e), showing a mineral assemblage of Amp + Bt + Pl + Kfs + Mag + Opx + Qz. The temperature range of the mineral assemblage is located below the solidus. The isopleths of $X_{An}$ in plagioclase

(0.26–0.3) constrain the P–T conditions of the M3 stage ~680–730 °C/3.6–5 kbar (yellow area in Figure 3b).

It can be seen that two metamorphic stages were determined from the peak temperature stage M2 via a transitional M2′ to a late retrograde M3 cooling stage. The derived peak temperatures reach UHT metamorphic conditions.

*5.2. Ti-in-Quartz Geothermometry*

In order to independently verify the P–T conditions derived from P–T pseudosection modeling, the quartz that is in equilibrium with mesoperthite/antiperthite or in the vein that could have also preserved related high-grade metamorphic information (Figure 2d,e) was applied for Ti-in-quartz thermometry. Therefore, a mineral reaction that essentially is reflected by the solubility of Ti-in-quartz [38,39] was used to track the temperatures of the studied rock. Since Ti-in-quartz is known to diffuse during retrograde metamorphism, it is often difficult to accurately determine the peak temperature conditions. From the results presented in Figure 3b, it becomes clear that using the data with the highest amounts of Ti-in-quartz, a temperature of about 950 °C can be derived which is already in the UHT stage and almost mirrors the peak temperature obtained from two-feldspar thermometry (see the next chapter).

*5.3. Two-Feldspar Geothermometry*

In order to reveal the original hypersolvus feldspar composition, two-feldspar thermometry was applied to derive peak metamorphic conditions by integrating perthite, antiperthite, and mesoperthite textures (Figure 2c–e) [23,40–42]. Previous studies have proposed different calculation models applying the two-feldspar geothermometer (Figure 4a–b). The models of Fuhrman and Lindsley and Benisek et al. [43,44] have been documented to be very robust for estimating temperatures, and thus are chosen for the current study. Using Photoshop software, we first preprocessed related BSE photographs of four mesoperthite/antiperthite domains, three of which are shown in Figure 4c–f. The area percentages of K-feldspar lamellae and host plagioclase were calculated according to the different color coding, using the respective pixel ratios. More information on the application of two-feldspar geothermometry to granulites is given by [41]. In the felsic-rich domain of the studied magnetite ore, several groups of antiperthite/mesoperthite microstructures were observed (Table S4) and the corresponding re-integrated ternary feldspar compositions are presented in Table S5 and Figure 4a,b. The peak temperature of the M2 stage calculated with a two-feldspar geothermometer is independent of pressure. Since the Opx at the M2 stage should be associated with plagioclase, which however is present only before the formation of the antiperthite/mesoperthite exsolution texture, the Opx in M2 stage is no longer preserved and the pressure of the M2 stage, therefore, cannot be constrained. Therefore, the pressure of M2 stage is taken from that of M2′ stage and represented the lowest pressure of M2 stage. Therefore, the maximum P–T conditions that the studied BIF iron ore has experienced are upgraded to a range of 1042–1061 °C/5 kbar and 1045–1080 °C/7 kbar, as indicated by the pink area in Figure 3b.

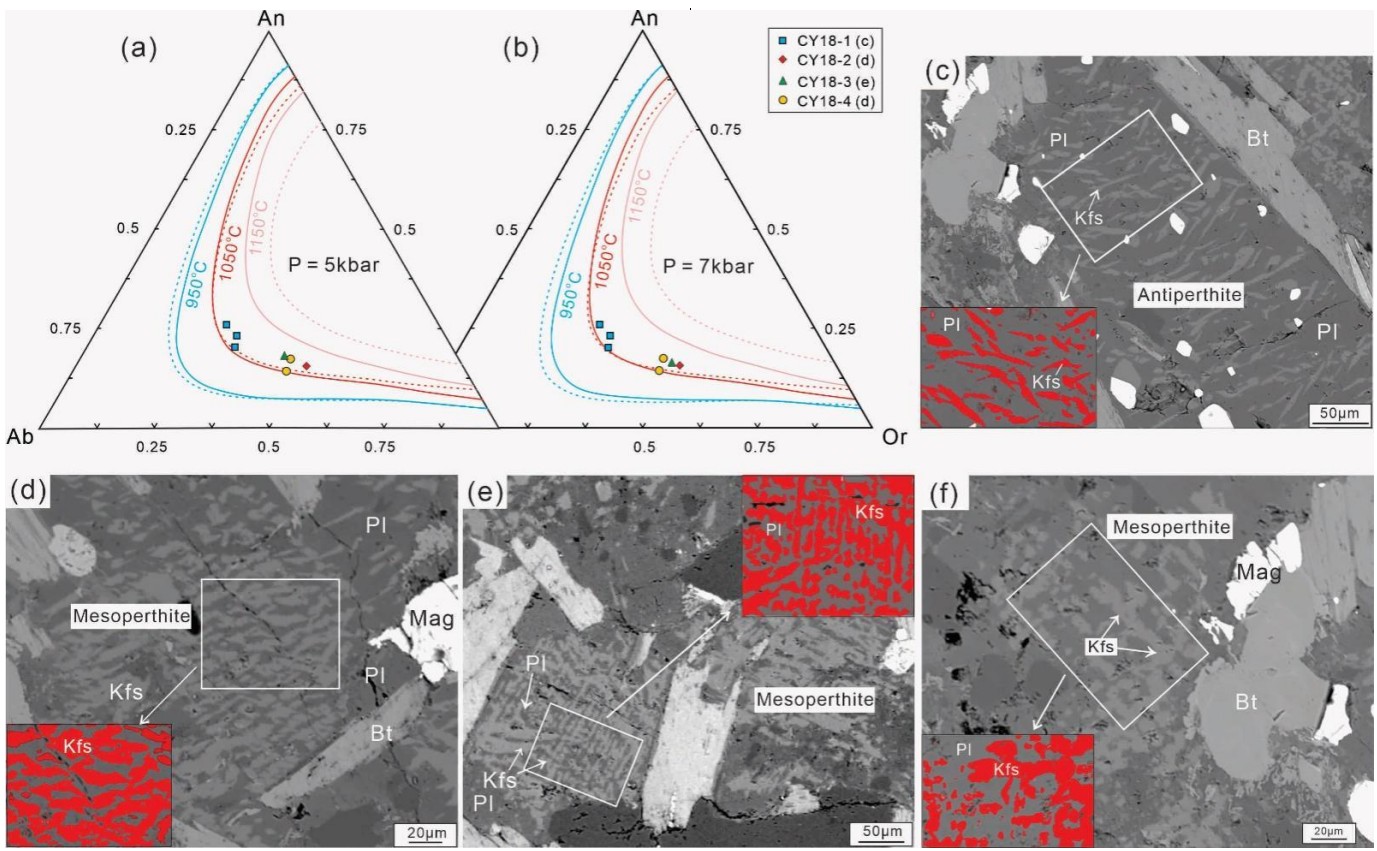

**Figure 4.** (**a**,**b**) Ternary diagrams of the reintegrated ternary feldspar compositions at 5 kbar and 7 kbar, respectively, where solid lines stand for the model of [43] and dashed lines for [44]; (**c**–**f**) or CY18-1–CY18-4 BSE photographs of antiperthite and mesoperthite texture in the BIF iron ore from the Zhengjiapo mine.

## 6. Discussion

### 6.1. Metamorphism of BIF IRON Ore

BIF iron deposits account for the majority of the world's iron resources and have been found in the Archean (>2.5 Ga), Paleoproterozoic Siderian (2.5~2.3 Ga), Orosirian (2.05~1.8 Ga), and Neoproterozoic Nanhua period (0.8~0.68 Ga). It formed on a large scale in the Neoarchean and Paleoproterozoic. After the Mesoproterozoic, the formation of BIF iron ore rarely occurred and only a small amount of BIF iron ore was formed during the "Snowball Earth Event" period about 800 Ma ago [1,19,45–47].

From Archean to Early Proterozoic, Fe was mainly derived from abiotic shallow sea hydrothermal fluid, which was mixed with volcanic material or with terrigenous detrital near the continent. Both endogenous and exogenous materials provide the necessary conditions for iron deposition; oxidation of $Fe^{2+}$ mainly took place by anoxic photosynthetic bacteria [47,48]. There may have been a variety of organic oxidation mechanisms in the ocean during this period, such as blue-green algae photosynthesis, anaerobic photosynthetic oxidation, and ultraviolet radiation oxidation, which converted dissolved $Fe^{2+}$ into $Fe^{3+}$ and then deposited solid, for instance, the iron-rich mineral $Fe(OH)_3$ [48,49]. Magnetite can be formed by reducing $Fe(OH)_3$ in seawater through microbial dissimilatory iron reduction [49].

BIF iron mineralization is spatially and temporally related to the orogenic Great Oxidation Event (GOE). Chen [45] has noticed that the properties of the lithosphere surface, atmosphere, hydrosphere, and biosphere have undergone fundamental changes at about 2.3 Ga. Based on studies of the so-called Lomagundi event, Melezhik et al. [50] proposed the correlation diagram considering early great oxidation events and corresponding

metallogenic events. In recent years, the spectrum of secondary events has been further improved [45], which is represented by a two-stage development. The early stage is hydrosphere oxidation (2.5–2.3 Ga) represented by the precipitation of the BIFs; the second-stage is atmospheric oxygenation during 2.3–2.05 Ga, indicated by sediments of thick carbonate strata with positive $\delta^{13}C$ carb excursion (Lomagundi Event).

After having formed in an Archean or Paleoproterozoic sedimentary basin, most BIFs underwent greenschist to amphibolite metamorphism and deformation during the Proterozoic orogenic events [1,6,51] and were then further modified by epigenic processes of later weathering [47,52].

Most known mineralizations of high-grade BIF iron ore occur after peak metamorphic conditions were reached; therefore, metamorphic fluid is considered to have no effect or little significance on iron ore grade improvement [51,53]. As mentioned above, iron mineralization is spatially and temporally related to orogenic events with extensive links from greenschist facies to amphibolite facies, even granulite facies metamorphism. For example, the BIF iron ore deposits in the Quadrilatero Ferrifero of Brazil underwent greenschist to amphibolite facies metamorphism [54]; several occurrences of magnetite-hematite ore from the Yilgarn Craton experienced lower greenschist to lower amphibolite facies metamorphism in [52,55]; banded iron formations in Um Nar, central eastern desert of Egypt, occur with intercalated greenschist facies metamorphic of volcaniclastic and epiclastic rocks in the ophiolitic-island arc system of the Arabo-Nubian [56] and around Mettupalayam within the Bhavani Suture Zone, South India granulite facies metamorphized BIF composed of magnetite and quartz occurs in association with meta-volcaniclastic rocks [57]. These Archean and Proterozoic BIF deposits are usually controlled by strike-slip fault zones or metamorphosed during different further orogenic processes [53].

Although not very common, a few BIF iron ores in orogenic belts with records of UHT metamorphism have been preserved. As important examples, UHT conditions can be recorded by two mineral assemblages, which refer to a clinopyroxene-orthopyroxene assemblage or their exsolution textures [14,15,58] and the existence of antiperthite, perthite, and mesoperthite in the BIF or associated metapelite [8]. Geothermometry using these two individual mineral assemblages or their combination with phase equilibria modeling showed that the corresponding metamorphic temperatures of all of them exceed 1000 °C (Figure 5a). It has been proposed that the UHT metamorphism of BIF or associated rocks occurred in a tectonic model involving decoupled break-off of the oceanic plate and plate subduction processes [58], or beneath a thickened continental crust during this period of Earth evolution [14,15]. Previous studies indicated that the rocks of the Zhengjiapo BIF iron deposit experienced metamorphic conditions exceeding 636 °C [1,6] and that the adjacent metasediments of the iron ores have experienced minimum P–T conditions of 810–830 °C and 6.3–6.9 kbar [21]. The current study is the first one to identify UHT metamorphism of BIF iron ores in the NCC; the related peak metamorphic overprint occurred at 1045–1080 °C at mid-crustal levels equaling pressures of 5.0–6.8 kbar.

Why is there no UHT metamorphic assemblage found in previous studies on the metapelite of the country rock of BIF mine? We think that the Zhengjiapo BIF iron ore is relatively dry and mainly composed of anhydrous minerals, such as magnetite, quartz, and feldspar. Due to its high melting temperature, close to crustal dry solidus (1050–1100 °C), and undergoing slow retrograde metamorphism, it is able to preserve UHT information [5,13,59]. In contrast, the adjacent amphibole gneiss is relatively wet since it contains more hydrous minerals, such as hornblende and biotite, and the melting temperature is relatively low; however, UHT assemblages may have existed which may have disappeared during retrograde metamorphism.

The studies on metamorphic BIF iron ore from the NCC show that the pressure conditions achieved are not very high on a global scale, generally in the range of low to medium pressures (Figure 5a). Although many BIF iron deposits have undergone metamorphism from amphibolite facies to granulite facies, most of them have not undergone deep subduction, but occur in an island arc accretive complex related to subduction [12,58,60–62] or in a

suprasubduction zone [5]. For example, in the late Neoarchean to early Paleoproterozoic Craton of the southern Indian Peninsula, there is a set of an arc-accretionary complex consisting of BIF, metasediment, metatuff, amphibolite, garnet-kyanite schist, metagabbro, pyroxenite, and charnockite. The pyroxene-rich domains of pyroxenite were formed at 900–1000 °C/1–1.2 GPa, whereas the garnet and clinopyroxene-rich domains of garnet clinopyroxenite show similar UHT temperature (900–1000 °C) and higher pressures of ~1.8~2 GPa [58]. The tectonic model of oceanic plate subduction refers to an initiation by break-off of the oceanic plate along the southern flanks of the Dharwar Craton [57]. The BIF iron ore bodies in the Yinshan block of the NCC are interlayered with plagioclase amphibolite in the greenstone belt, and their formation is linked to seafloor volcanism and hydrothermal activity in a ridge subduction related slab window [60]. High-temperature meta-ironstone, associated with pyroxenites and spinel-lherzolites from East Tonagh Island, Enderby Land, Antarctica, preserved exsolution textures that indicate the former coexistence of metamorphic pigeonite and ferroaugite at temperatures more than 980 °C and pressures ~7 kbar [12]. BIFs associated with ophiolite succession consisting of serpentinite, lherzolite, olivine clinopyroxenite, websterite, gabbro, dolerite, diorite, monzonite, metabasalt, and trondhjemite in the Miyun complex of the NCC were found to be derived from suprasubduction zone in the Paleoproterozoic and shown to have experienced subduction-, accretion-, and collision processes [5]. Experimental study has shown that the carboniferous BIF can melt, and that all carbonate minerals left the system at around 6 GPa during subduction under high-Archean geothermal gradient [53]. Fe-rich carbonatites should ascent but stagnate gravitationally near the slab/mantle interface until Fe-Mg exchange and partial reduction reaction with the mantle occur. It can be seen that the density of the BIF is greater than that of the subduction plate or mantle material due to the high iron content, in order that the BIF rarely returns to the surface after subduction [62]. Based on gravity calculations, BIF is likely to stabilize in the ultra-low velocity zone at the Earth's core-mantle boundary [62].

### *6.2. P–T Conditions and Geological Implication*

Types of BIF iron deposits are widely distributed in greenstone belts of the NCC, such as at eastern Hebei, Wutai, western Shandong, and Anshan-Benxi, etc. Formation ages of these deposits were mostly in Neoarchean from 2.62 Ga to 2.50 Ga, whereas the metamorphic overprint occurred from Late Neoarchean to early Paleoproterozoic (~2.53–2.44 Ga) [2,63,64]. These BIF iron deposits were stratigraphically interlayered with submarine-emplaced volcanic rocks and underwent greenschist to granulite facies metamorphism due to multiple tectonic events during the amalgamation of the various Archean micro-blocks [18,51], or related to a Neoarchean subduction of an oceanic slab and subsequent accretion- and collision-processes at the Archean-Proterozoic transition [2,18,25,63]. However, the Zhengjiapo BIF iron deposit in the Changyi metallogenic belt discussed in this paper occurs in the Paleoproterozoic orogenic belt of the North China Craton.

The JLJB is one of the three Paleoproterozoic orogenic belts, which is located in the Eastern Block of the NCC (Figure 1a). Its Precambrian basement is metamorphosed and deformed and includes Meso- and Neoarchean-rocks, which mainly are TTG gneisses and associated supracrustal rocks or metapelites. Metapelites are representing large volumes of metamorphic meta-sedimentary-volcanic successions in the Ji'an and Laoling Groups in the southern Jilin province, the North and South Liaohe Groups in the Liaoning Peninsula, the Fenzishan and Jingshang Groups in the Jiaobei terrane, and the Wuhe Group in the Anhui Province from NE to SW of the JLJB [16,17,65].

Most of the Archean basement rocks in the JLJB are derived from magmatic emplacement at ~2.6–2.5 Ga, while only a few were formed at 2.8–2.7 Ga [3,17,66,67]. The superimposition of ~2.5 Ga tectono-thermal events in the TTG gneiss of the Jiao-Liao-Ji orogenic belt and other terranes in the NCC is characterized by different magmatic and metamorphic events [3,68,69]. After ~2.5 Ga, the TTG gneisses recorded Paleoproterozoic HP metamorphic ages of ~1.9–1.8 Ga in JLJB [68–71].

Supracrustal rocks or metapelites from the Fenzishan and Jingshan Groups of the Jiaobei Terrane, northern and southern Liaohe Groups and the Ji'an and Laoling Groups were deposited within the Paleoproterozoic JLJB that was developed during the period ~ 2.3–2.0 Ga on an Achaean basement [3,6,19,54,68,72]. In addition, abundant granitoid and mafic intrusions were developed during 2.2–2.1 Ga in the JLJB. The deposition of the Fenzishan and Jingshan supracrustal rocks was accompanied by the emplacement of A-type granite as well as gabbroic and doleritic dykes in an extensional setting [6,20,70]. Subsequently, these rocks underwent amphibolite to granulite facies conditions, some of which even reached UHT granulitic facies during 1.92–1.76 Ga, resulting from subduction and collision processes that eventually led to the closure of the rift basin [65,67,71,73–77].

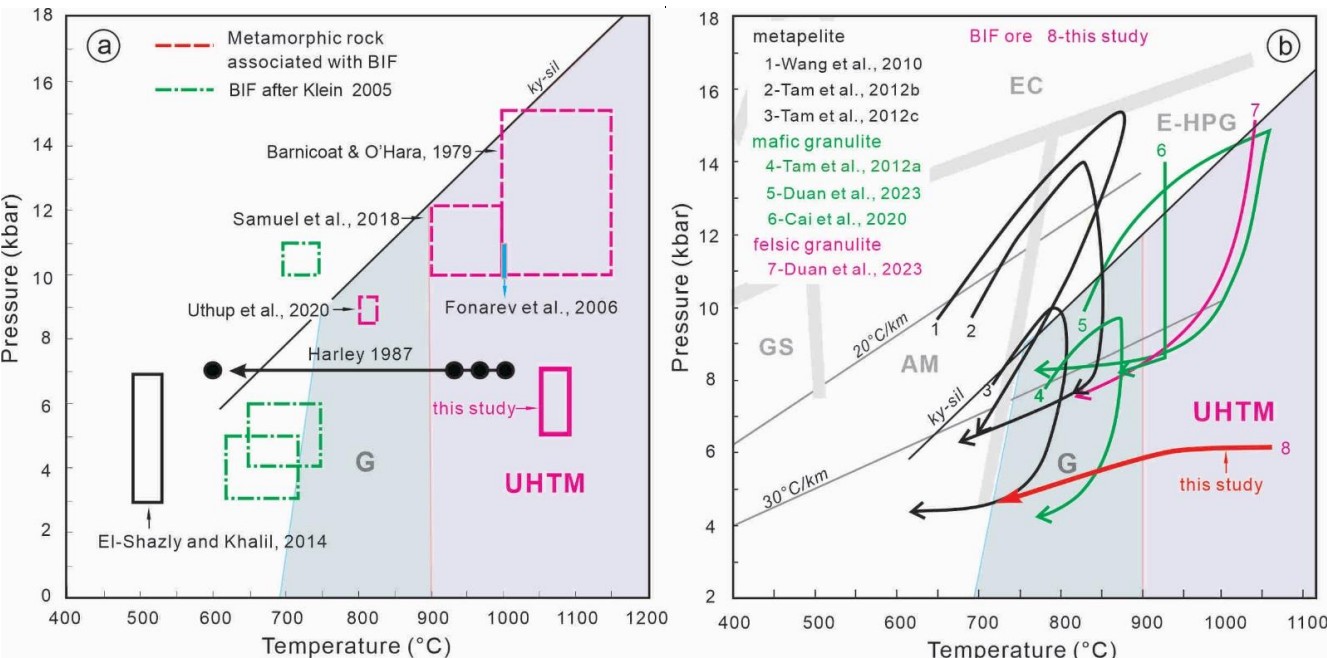

**Figure 5.** (**a**) Evaluations of the P–T stability fields of represented metamorphic BIF in the world (modified after [47]) [12,14,15,56–58]; (**b**) HT-UHT metamorphic records from the mafic and granitic granulites in the JLJB. All P–T trajectory data in Figure 5b were taken from the rocks in the Jingshan Group, while samples in this study were taken from the Fenzishan Group [71,73–77].

The P–T evolution of pelitic, mafic, and granitic granulites in the JLJB has been studied extensively in the last decade (e.g., [71,73–77], Figure 5b). For pelitic granulites in the Jingshan Group of Jiaobei terrane, the peak P–T conditions of high pressure (HP) and high temperature (HT) are estimated by using traditional geothermobarometers or phase equilibrium method, and clockwise P–T trajectories were derived [73,75,76]. The mineral assemblages of the peak stages are garnet + kyanite + biotite + plagioclase + K-feldspar + quartz + ilmenite + rutile + liq for HP pelitic granulite, and garnet + sillimanite + biotite + plagioclase + K-feldspar + quartz + ilmenite + liq for MP pelitic granulite. However, to date, the UHT metamorphic conditions of pelitic granulites in the JLJB have not been reported. The mafic granulites in the eastern Liaoning and Jiaobei terrane have been studied by means of phase equilibrium method. In addition to the medium pressure (MP) to high temperature metamorphism (HT) [74], UHT metamorphism conditions have been reported in recent years [71,77]. Their mineral assemblages of peak metamorphic stage are reflected by garnet + clinopyroxene + plagioclase + quartz ± rutile + liq and garnet + clinopyroxene + plagioclase + K-feldspar ± amphibole ± quartz ± rutile + liq, respectively. The obtained P–T conditions are ~14–15 kbar and 920–1073 °C for the peak metamorphic stage, and P–T trajectories are clockwise [60,65]. Furthermore, the felsic granulites of the Jiaobei terrane have recorded UHT metamorphism similar to that of mafic granulites, with

a mineral assemblage of garnet + plagioclase + K-feldspar + quartz + ilmenite + liq and corresponding peak metamorphic conditions ranging from 14 to 16.5 kbar and 985 to 1077 °C [77]. The peak HP-UHT stage obtained from metamorphic zircon domains in felsic and mafic granulites is constrained by an age of ~1950–1900 Ma, which was a response to a Paleoproterozoic continent–continent subduction and collision [71,76]. The P–T path 8 of this study in Figure 5b comes from the BIF ore of the Zhengjiapo BIF mine in the Fenzishan Group, which only records an incomplete track, representing the cooling process from the peak UHT metamorphism. Note that the BIF mine in the Fenzishan Group is located in the northern belt of the JLJB, and thus the geological implications of the P–T path in this study cannot be directly compared with other P–T trajectories from the southern belt of the JLJB listed in Figure 5b. However, they are all the products of the Paleoproterozoic Jiao-Liao-Ji orogenic belt and have experienced the basin formation, subduction, and collision orogenic processes together. The P–T thermodynamic processes of the Fenzishan and Jingshan Groups in the JLJB may be different. In terms of the formation ages of HT-UHT, both groups of rocks were formed in ~1.95–1.90 Ga, indicating that the HT-UHT metamorphic event is related to the continental subduction-collision processes [7,71,74,77].

The Changyi BIF iron metallogenic belt is part of the supracrustal sequences of the Fenzishan Group in the Jiaobei terrane, which is deposited at the margin of continental rift basin at ~2.2 Ga [19], and thus the JLJB in the Eastern Block of the NCC experienced a metamorphism in the Paleoproterozoic at 1.95–1.80 Ga [1,17,19,20]. The deposition of the Fenzishan supracrustal rocks was accompanied by A-type granitic intrusion, indicating that the Changyi BIFs, including the Zhengjiapo BIF mine, formed under the tectonic background of continental rift, i.e., in the JLJB [1,20,70]. The new data of this study bear fundamental geodynamic consequences, since, based on the new P–T estimates of the BIF iron ore, it demonstrates that the Zhengjiapo BIF iron mine of the Changyi metallogenic belt has experienced UHT metamorphic conditions at ~1.9 Ga, as indicated by apatite U-Pb dating [21], which is consistent with the regional UHT metamorphic event discussed above (Figure 5b). On the basis of coexistence of antiperthite and mesoperthite and phase equilibrium modeling of the ore sample, the estimated peak P–T conditions reach up to 1045–1080 °C/~5.0–6.8 kbar, which could be a result of the collision-related process of the JLJB. Since the mineral assemblages of BIFs worldwide with magnetite, hematite, quartz, and chert are, in general, quite simple; therefore, any kind of metamorphic overprint cannot be easily identified, and may often have been overlooked. A focus on rare silicate minerals if present, such as feldspar, orthopyroxene, and amphibole in the current study, may also help in uncovering the possible UHT metamorphic grade in other occurrences.

## 7. Conclusions

1.  Specimens of BIF iron ore were collected as representative for granulite-facies metamorphic BIF of the Zhengjiapo iron mine in the Jiao-Liao-Ji Belt. Three stages of metamorphism were identified in the felsic-rich domain of the ore sample, with a peak metamorphic assemblage of Qz + Pl + Kfs + Opx + Mag + Liq (M2), a post-peak cooling assemblage of Bt + Amp + Opx+ Pl + Kfs + Qz + Mag (M3). The granulite-facies BIF is documented by the occurrence of antiperthite and mesoperthite that developed at the peak UHT metamorphic conditions.
2.  Ternary feldspar thermometry using re-integrated compositions of antiperthite and mesoperthite in the felsic domain of the BIF iron ore yields peak metamorphic temperatures of 1045–1080 °C and estimated pressures of 5.0–6.8 kbar. Thermobarometry combined with phase equilibria modeling revealed retrograde P–T conditions of ~680–730 °C/3.6–5 kbar during the process of cooling.
3.  The result of this study documents important details of the metamorphic conditions of the BIF iron ore in the JLJB and sheds new light on the metamorphic development of the NCC with respect to the derived HT to UHT granulite facies conditions. The granulite facies BIF ore of the Zhengjiapo iron mine is possible as a result of geo-

dynamic processes related to continental collision, followed by exhumation of the Paleoproterozoic JLJ orogenic belt.

**Supplementary Materials:** The following supporting information can be downloaded at: https://www.mdpi.com/article/10.3390/min13070980/s1, Table S1: The mineral compositions (wt.%) of assemblage used to calculate peak temperature conditions for sample 19CY18 in the Zhengjiapo BIF iron deposit of the Changyi district, Table S2: Other mineral compositions (wt.%) for the sample 19CY18 in the Zhengjiapo BIF iron deposit of Changyi district, Table S3: Major compositions (wt.%) of the BIF iron ore sample (19CY18), Table S4: Representative EMP analyses of antiperthite/mesoperthite in the BIF iron ore sample (19CY18), Table S5: Reintergrated compositions of feldspar with area proportion and chemical compositions of lamellae and host domains.

**Author Contributions:** Y.-R.C. contributed in terms of data analyzing, modeling, original draft, and writing; X.-P.L. designed the experiments, data evaluation, writing—review and editing, and supervision; Z.-S.L. contributed in terms of methodology, resources, and review; H.-P.S. and F.-M.K. contributed in terms of data evaluation, writing—review and editing, and supervision. Conceptualization, X.-P.L. and Y.-R.C.; methodology, Y.-R.C. and Z.-S.L.; formal analysis, Y.-R.C. and Z.-S.L.; investigation, X.-P.L., Y.-R.C. and F.-M.K.; writing—original draft preparation, Y.-R.C.; writing—review and editing, X.-P.L. and H.-P.S.; supervision, X.-P.L.; funding acquisition, X.-P.L. All authors have read and agreed to the published version of the manuscript.

**Funding:** The current study was supported by the National Natural Science Foundation of China (U1906207) and (U2244206); and by the Open Project of Key Laboratory of Gold Mineralization Processes and Resource Utilization Subordinated to the Ministry of Land and Resources and Key Laboratory of Metallogenic Geological Process and Resources Utilization in Shandong Province (KFKT202105).

**Data Availability Statement:** Data are contained within the Supplementary Materials.

**Acknowledgments:** We are grateful to the anonymous reviewers who helped in improving the manuscript. Our sincere thanks also goes to Chiara Groppo, Torino, for very valuable suggestions of an earlier version of this paper.

**Conflicts of Interest:** The authors declare no conflict of interest.

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
