# Peer review of "Antiperthite and Mesoperthite Exsolution Textures in the Zhengjiapo BIF, Changyi Metallogenic Belt, North China Craton: Evidence of UHT Metamorphic Overprint"

_minerals, doi:10.3390/min13070980_

Round 1

Reviewer 1 Report (New Reviewer)

To my knowledge, this is the first time to report UHT metamorphism of the Zhengjiapo BIF, the Changyi metallogenic belt, Eastern Block of North China Craton. Based on detailed micropetrographic observation, the authors did pseudosection modeling and geothermometry research, and conclude that the iron rock experienced UHT metamorphic event in the Paleoproterozoic. The data are solid and the computation is valid.

Specific comments:

(a) About syntax. I suggest that the English be polished throughout. E.g., Line 23, “is” should be corrected as “was”; Line 104, “consist” should be corrected as “consisting”; Line 116, “and” can be corrected as “or”, ……

(b) I suggest that the authors use single rather than plural form for mineral or rock name, such as using “antiperthite” rather than “antiperthites”.

(c) Please specify that why do not the surrounding rocks, such as garnet amphibolite, metapelite, record UHT metamorphic clues?

The English can be polished

Author Response

Respons to Reviewer 1#

Comments and Suggestions for Authors

To my knowledge, this is the first time to report UHT metamorphism of the Zhengjiapo BIF, the Changyi metallogenic belt, Eastern Block of North China Craton. Based on detailed micropetrographic observation, the authors did pseudosection modeling and geothermometry research, and conclude that the iron rock experienced UHT metamorphic event in the Paleoproterozoic. The data are solid and the computation is valid.

Reply: Thanks for positive comments.

Specific comments:

(a) About syntax. I suggest that the English be polished throughout. E.g., Line 23, “is” should be corrected as “was”; Line 104, “consist” should be corrected as “consisting”; Line 116, “and” can be corrected as “or”, ……

Reply: All problems have been corrected, and the English been polished throughout.

(b) I suggest that the authors use single rather than plural form for mineral or rock name, such as using “antiperthite” rather than “antiperthites”.

Reply: all names of mineral or rock are revised to singular form.

(c) Please specify that why do not the surrounding rocks, such as garnet amphibolite, metapelite, record UHT metamorphic clues?

Reply: Why is there no UHT metamorphic assemblage found in previous studies on the metapelite of the country rock of BIF mine? We think that the Zhengjiapo BIF iron ore is relatively dry and mainly composed of anhydrous minerals such as magnetite, quartz and feldspar. Due to its high melting temperature, close to crustal dry solidus (1050−1100 °C), and undergoing slow retrograde metamorphism, it is easier to preserve UHT information (Santosh et al., 2020; Wang B et al., 2020, 2023). In contrast, the adjacent amphibole gneiss is relatively wet because it contains more hydrous minerals such as hornblende and biotite, and the melting temperature is relatively low, so it is difficult to reach the UHT metamorphic conditions, or even if UHT assemblage has existed, it is easy to disappear in the faster retrograde metamorphic process quickly. We add this paragraph in the text. Please see lines 461-470.

Best Regards,

Xu-Ping Li

Reviewer 2 Report (New Reviewer)

Dear authors,

The authors present new metamorphic PT conditions in the Zhengjiapo BIF. The main objective of the study is to determine the metamorphic conditions of the BIF. The authors suggest the existence of ultrahigh temperature metamorphism, and the involvement of the BIF line the collision related tectonic process.

However, the major problem with this publication is a not complete state of the art of the geological background. There is no information about the age of the Changyi bif and the age of the metamorphism. And, finally line 517 we discover a publication [70] on chronology, geochemistry and isotopes of the Changyi BIF in the same Jinshan Group. Moreover, this publication suggests an age for the amphibole metamorphism. The authors do not present this age in the publication (why?), they prefer to quote an unpublished age, which is comparable.

The geological context section must be rewritten and must imperatively present all existing data on BIFs of the Changyi metallogenic belt to have a better overview and to better appreciate the new results of this publication.

The new path PT conditions obtained in figure 3 should be introduced in figure 5b. And explain how to reconcile this new path with the paths obtained on metapelites, mafic and felsic granulites.

The geological contexts characterized by UHT should also be described more precisely. Are these UHT metamorphic contexts more frequently encountered in the Archean or Proterozoic? Are they characteristic of Archean orogens or modern-style orogens? What are the implications for the characterization of Paleoproterozoic orogens?

Best regards

Author Response

Response to Reviewer 2#

Comments and Suggestions for Authors

Dear authors,

The authors present new metamorphic PT conditions in the Zhengjiapo BIF. The main objective of the study is to determine the metamorphic conditions of the BIF. The authors suggest the existence of ultrahigh temperature metamorphism, and the involvement of the BIF line the collision related tectonic process.

However, the major problem with this publication is a not complete state of the art of the geological background. There is no information about the age of the Changyi bif and the age of the metamorphism. And, finally line 517 we discover a publication [70] on chronology, geochemistry and isotopes of the Changyi BIF in the same Jinshan Group. Moreover, this publication suggests an age for the amphibole metamorphism. The authors do not present this age in the publication (why?), they prefer to quote an unpublished age, which is comparable.

Reply: It is not really true, we not only cited age information [70] ( Lan, T.G. et al., 2014b), but also cite all represented references related to U-Pb dating results of the Changyi BIF ore deposit [1, 6, 64] (Lan, T.G. et al., 2014a, Wang HC et al., 2015, Lan, T.G. et al., 2015) in the places where different ages involved in the previous MS (see previous MS Lines 499-502, 538, 541). As reviewer suggested, we added the summary of previous dating information for the Changyi BIF studies in revised MS, and added relevant content in the geological background. Please see lines 113-136.

This paper mainly studies the UHT metamorphism found in BIF iron ore of the Changyi BIF, without extensive discussion on the age of metamorphism. Another paper we submitted specifically discusses the detrital zircon ages, including metamorphic age, of amphibole gneiss associated with BIF iron ore. In addition, those dating samples in the submitted paper is not from this study, so we only cite age here.

The geological context section must be rewritten and must imperatively present all existing data on BIFs of the Changyi metallogenic belt to have a better overview and to better appreciate the new results of this publication.

Reply: see above question.

The new path PT conditions obtained in figure 3 should be introduced in figure 5b. And explain how to reconcile this new path with the paths obtained on metapelites, mafic and felsic granulites.

Reply: the new P-T path of this study is added in figure 5b, and give an explanation in the text. Please see lines 564-569.   

The geological contexts characterized by UHT should also be described more precisely. Are these UHT metamorphic contexts more frequently encountered in the Archean or Proterozoic? Are they characteristic of Archean orogens or modern-style orogens? What are the implications for the characterization of Paleoproterozoic orogens?

Reply: we added UHT related information in the introduction section, please see lines 56-64 and lines 400-405. In 6.2. P-T conditions and Geological implication, combined with current available literature, we discuss in detail the formation, metamorphism and evolution history of the JLJB, including Changyi BIF iron ore. We concluded that the UHT metamorphism of Changyi BIF is consistent with the regional UHT metamorphism, which could be resulted from the collision-related process of the JLJB.

       We first present the Neoarchaean BIF distributed in the greenstone belt of the North China Craton, which occurred from 2.62 Ga to 2.50 Ga in lines 507-527;

We discuss the Archean basement of JLJB, its age spectrum and geological implication in lines 526-531;

Then, the BIF in the Changyi metallogenic belt, which originated in the Paleoproterozoic JLJB of the North China Craton, is discussed. Furthermore, petrographic component and evolution of JLJB are discussed in lines 532-541;  

We compare the P-T path of the Zhengjiapo BIF with the UHT metamorphic rocks from the JLJB, and discuss their geological implication in lines 542-569;

Changyi BIF is hosted in the Fenzishan Group; we discuss supracrustal rock or metapelite of the Fenzishan and Jingshan Groups, their rock series, age spectrum and geological implication, the formation and development of the JLJB. Finally, we discuss the formation, metamorphism and geological implication of Zhengjiapo BIF in lines 570-588;

Best regards

Xu-Ping Li

Round 2

Reviewer 2 Report (New Reviewer)

Dear Authors,

Corrections have been made to the first version of the manuscript. They improve understanding of the manuscript.

It would be great if the authors could make all the spelling corrections in English between the two versions. There are still a lot of poorly orthographic Archean or Neoarchean terms. These terms were pointed out in the first review.

There are still many places where the term Archean is misspelled. These have not been corrected (lines 33, 65, 86, 150, 152, 163, 425, 447).

My main comment concerns the comparison with existing P-T paths. The authors see the difference but don't explain it. They get away with a pirouette, claiming that the different data do not come from the same area.

This is the case for some of the data, but not for the paths defined by Duan et al 2023, which come from the same sector.

This publication                                                                  Duan et al 2023

According to the authors, the P-T paths cannot be directly compared. This does not prevent the authors from asserting (lines 602-607), that these new P-T data are consistent with the regional data. So, 10 lines later, they can be directly compared...

In this version, the authors continue to refer to unpublished data. And the number of references to unpublished data has increased compared with the previous version. I suggest that these references to unpublished data be removed.

If this data is essential to understanding the publication, it must be included in the publication.

Regards,

All the spelling corrections indicated in the first review have not been realized.

Author Response

Reply to Reviewer Comments

Dear Authors,

Corrections have been made to the first version of the manuscript. They improve understanding of the manuscript.

It would be great if the authors could make all the spelling corrections in English between the two versions. There are still a lot of poorly orthographic Archean or Neoarchean terms. These terms were pointed out in the first review.

Reply, we checked the manuscript carefully and made the respective corrections required.

There are still many places where the term Archean is misspelled. These have not been corrected (lines 33, 65, 86, 150, 152, 163, 425, 447).

Reply, all relevant spellings throughout the text are corrected.

My main comment concerns the comparison with existing P-T paths. The authors see the difference but don't explain it. They get away with a pirouette, claiming that the different data do not come from the same area.

Reply, Please see below. In the title of Figure 5b. We added the explanation” All P-T trajectory data in Figure 5b were taken from the rocks in the Jingshan Group, while samples in this study were taken from the Fenzishan Group.”

This is the case for some of the data, but not for the paths defined by Duan et al 2023, which come from the same sector.

Reply, This is not true. First of all, the two maps the reviewer has attached are identical. However, this map is only the one of our figures in current manuscript, but NOT the one in the Duan et al (2023)-paper . The two respective correct maps are given below and document (a) that they are different and (b), even more important, that the rocks being studied came from two completely different places nearly 100 kilometers apart  (see violet circles with arrows below); the studied samples were taken from the Fenzishan Group, while Duan et al.(2023) 's research samples were taken from Jingshan Group.

   Duan et al. (2023)                                         Chen et al. (current manuscript)

According to the authors, the P-T paths cannot be directly compared. This does not prevent the authors from asserting (lines 602-607), that these new P-T data are consistent with the regional data. So, 10 lines later, they can be directly compared...

Reply, As said above that all P-T trajectory data in Figure 5b were taken from the rocks in the Jingshan Group, while samples in this study were taken from the Fenzishan Group. Tectonically, the Paleoproterozoic Fenzishan and Jingshan Groups directly overlie the Archea-an-Paleoproterozoic basement rocks with the ductile shear zone developing along the contact boundary…see lines 106-108. We added a paragraph to explain why the P-T paths of the two Groups cannot be directly compared, but they can share the same geothermal events in the same orogenic belt. See line 477-478.

In this version, the authors continue to refer to unpublished data. And the number of references to unpublished data has increased compared with the previous version. I suggest that these references to unpublished data be removed.

Reply, we added the reference [21], which is Chen’s master's thesis.

If this data is essential to understanding the publication, it must be included in the publication.

Reply, this paper mainly studies the UHT metamorphism found in BIF iron ore of the Changyi BIF, without extensive discussion on the age of metamorphism. We do not consider it necessary to include age data for the following reasons:

  • This age of ~1.9 Ga is not first  published in the JLJB, and  it is  well accepted  representing  a subduction-cllision  processes in the JLJB (e.g. Tam et al., 2012a; Li Z et al., 2017; Xu et al., 2018a,b; Zou et al., 2018; Cai et al., 2020; Duan et al., 2023); 
  • Lan et al. (2014a) reported that the Changyi BIF underwent amphibolite facies metamorphism at about 1864 Ma, which is consistent with widely recorded age of 1.9 Ga from the Paleoproterozoic sequences in Jiaobei terrain and adjacent areas (e.g. Zhao et al., 2005; 2012; Li & Zhao, 2007; Zhai & Santosh, 2011; Tam et al., 2012a; Liu et al., 2019; Cai et al., 2020; Duan et al., 2023).
  • We didn’t list this age as an individual conclusion, but only cite this age when discussing the possible geological significance of the UHT metamorphic event.
  • Many papers have been published, focusing on metamorphism, mineralization, or geochemistry, but partly without analyzing the data, citing only references or unpublished data (e.g. Tam et al., 2012a; Hein, 2002; Hart et al., 2002; Homam, 2015; Zarasvandi et al., 2019; Warr et al., 2021; Cunningham et al., 2023).

Addition references:

Li, Z.; Chen, B.; Wei, CJ. 2017. Is the Paleoproterozoic Jiao-Liao-Ji Belt (North China Craton) a rift? International Journal of Earth Sciences 106(1), 355-375.

Xu, W., Liu, F.L., Tian, Z.H., Liu, L.S., Ji, L., Dong, Y.S., 2018a. Source and petrogenesis of Paleoproterozoic meta-mafic rocks intruding into the North Liaohe Group: implications for back-arc extension prior to the formation of the Jiao-Liao-Ji Belt, North China Craton. Precambrian Res. 307, 66–81.

Xu, W., Liu, F.L., Santosh, M., Liu, P.H., Tian, Z.H., Dong, Y.S., 2018b. Constraints of mafic rocks on a Paleoproterozoic back-arc in the Jiao-Liao-Ji Belt, North China Craton. J. Asian Earth Sci. 166, 195–209.

Zou, Y.; Zhai, MG.; Santosh, M.; Zhou, LG.; Zhao, L.; Lu, JS.; Liu, B.; Shan, HX. 2018. Contrasting P-T-t paths from a Paleoproterozoic metamorphic orogen: Petrology, phase equilibria, zircon and monazite geochronology of metapelites from the Jiao-Liao-Ji belt, North China Craton. Precambrian Research 311, 74-97

Liu, F.; Liu, L.; Cai, J.; Liu, P.; Wang, F.; Liu, C.; Liu, J. 2019. A widespread Paleoproterozoic partial melting event within the Jiao-Liao-Ji Belt, North China Craton: Zircon U-Pb dating of granitic leucosomes within pelitic granulites and its tectonic implications. Precambrian Researc 326,155-173.

Hein, K.A.A. 2002. Geology of the Ranger Uranium Mine,Northern Territory,Australia: structural constraints on the timing of uranium emplacement. Ore Geology Reviews 20(3-4), 83-108 doi:10.1016/S0169-1368(02)00054-9

Hart, C.J.R.; Goldfarb, R.J.; Qiu, Y.M.; Snee, L.; Miller, L.D.; Miller, M.L. 2002. Gold deposits of the northern margin of the North China Craton: multiple late Paleozoic-Mesozoic mineralizing events. Mineralium Deposita 37(3-4), 326-351 doi:10.1007/s00126-001-0239-2

Homam, S.M. 2015. Petrology and geochemistry of Late Proterozoic hornblende gabbros from southeast of Fariman, Khorasan Razavi province, Iran. Journal of Economic Geology 7(1), 91-109

Zarasvandi, A.; Rezaei, M.; Tashi, M.; Fereydouni, Z.; Saed, M. 2019. Comparison of geochemistry and porphyry copper mineralization efficiency in granitoids of the Sanandaj-Sirjan and Urumieh-Dokhtar zones; using rare earth elements geochemistry. Journal of Economic Geology 11(1), 1-32 doi:10.22067/econg.v11i1.64479

Warr, O.; Giunta, T.; Onstott, T.; Kieft, T.; Harris, R.; Nisson, D.; Lollar, B.S. 2021. The role of low-temperature 18O exchange in the isotopic evolution of deep subsurface fluids. Chemical Geology 561, 120027 doi:10.1016/j.chemgeo.2020.120027

Cunningham, J.K.; Gómez-Fernández, F.; González-Menéndez, L.; Beard, A.D. 2023.  Black shales and mesozonal quartz vein-hosted Au: The Truchas Syncline, Spain and the Harlech Dome, Wales, a comparative study. Geological Journal 58(1), 85-107 doi:10.1002/gj.4581

Regards,

Comments on the Quality of English Language

Reply, as you can see from the improved manuscript, we have considerably polished the manuscript regarding English language.

All the spelling corrections indicated in the first review have not been realized.

Reply, we checked the manuscript carefully and made the respective corrections required.

Submission Date

26 April 2023

Date of this review

27 Jun 2023 08:21:34

Round 3

Reviewer 2 Report (New Reviewer)

Dear authors,

We appreciate the efforts made since the first version in terms of presentation quality and scientific soundness.

We accept in present form the manuscript.

Best regards

This manuscript is a resubmission of an earlier submission. The following is a list of the peer review reports and author responses from that submission.

Round 1

Reviewer 1 Report

Summary

This manuscript examines the anti-perthite and meso-perthitic textures in feldspar within BIF of the Zhengjiapo Mine in the Changyi Belt which forms part of the Jiao-Liao-Ji Belt (JLJB) of northeastern China. Using a sample of the BIF and determining peak and post-peak metamorphic assemblages, undertaking geochemistry of a homogenous leucocratic portion of the BIF, and doing mineral chemistry the P-T conditions the BIF were subjected to were determined. The study also makes use of two-feldspar geothermometry by visually determining the ratio of Na-feldspar to K-feldspar in the anti- and meso-perthitic feldspars and thereby determines the temperatures to which the BIF was subjected. These P-T conditions are used to highlight UHT conditions, higher than previous determinations of metamorphic conditions. This, along with age data not reported in this manuscript, but done as part of the study, help constrain the tectonothermal evolution of this portion, at least, of the JLJB.

The data used are appropriate to constrain the P-T conditions. The petrography, however, is poorly done and confusing and leaves the reader somewhat confused as to the mineral assemblages and what compositions belong with what. It’s also not clear how the authors determined the different mineral assemblages (M2’ and M3). Was it based on associations or textures? No geological setting is given and so it is difficult for readers to place the area or samples in a proper geological or tectonic context or even care why this methodology was done. This should be done more extensively upfront.

 The findings add to the literature to constraining the P-T conditions of the JLJB. The manuscript is short and concise. In fact, too short and concise and leaves out some details that will help the reader place the study in a greater geological context or appreciate its significance. There are numerous examples of grammatical errors which detracts from the work and the science. Generally, the manuscript draws appropriate conclusions and interpretations from the data.

General comments

The subject matter is of general interest to Minerals readers, particularly those in East Asia, but does have a global relevance with regards to the metamorphism of BIF ores and using various techniques to determine the P-T conditions of metamorphism.

The interpretation adds important information to the tectonic development of the JLJB but the reader should be given the geological context better. The title is appropriate to the study and its content.

The authors have gone for a short, concise manuscript but at the expense of giving information that would highlight the geological context and significance of the findings to not only the region but to other metamorphosed BIFs globally. A brief geological setting and overview is not given. Not all readers are familiar with the Jiao-Liao-Ji belt. The geology, overall setting, and tectonic, metamorphic and deformational history of the JLJB should be given so that the reader has an idea as to the geological context, when the JLJB formed, when the BIF was likely deposited and when metamorphic events occurred and what their tectonic or geodynamic setting have been postulated to have been. This would place the findings of this study in a better context and highlight its relevance to other metamorphosed BIFs around the globe. As such, currently as the manuscript is structured, the reader needs to gain this information piecemeal from reading other articles referenced throughout the manuscript. Many readers would not be bothered trying to chase down a number of other articles to gain an overview of the geological setting and relevance of the findings given in this study. It should, then, be briefly given in this manuscript up front. International readers may be frustrated by a lack of a proper geological context. As an example, groups are mentioned and given with no mention of their rock types, again. Short and concise is fine, but the reader needs to be given some information to make their reading worthwhile.

The pressures measured of 5.2-7.2 kbar are not consistent with collisional tectonics being too low. Collision would lead to thicker crust of 50 to 70 km thickness (~13 to 20 kbar)

The petrographic or sample description could be better organized and more descriptive detailing mineral assemblages. The description should be split into the ore-rich domains first and then the felsic-rich (or leucocratic) domains. Microperthite and antiperthite (strictly speaking these are textures) are assigned to the M2 metamorphic stage, but no reasoning or justification for this is given. How is it known that these minerals form part of the M2 peak metamorphic assemblage? A description of the metamorphic assemblages should be given and descriptions and explanations as to how these were determined from the textures, mineral assemblages and associations. Different textures and assemblages which indicate when these different minerals formed as part of differing metamorphic stages should be described better and in more detail. It would be helpful to clearly state what the M2 and M3 mineral assemblages are at the end of the petrographic description section as this relates to what comes later in the pseudosection modelling.

There was an error in reading off the pressures and temperatures determined from the pseudosection for M2’ conditions (Fig. 3b in the manuscript). This may influence which pressures were used to determine the temperatures using the two-feldspar geothermometer. As such, this may lead to some errors in the temperatures and pressures reported for the UHT metamorphism. The authors are requested to please check this.

The manuscript initially does not flow well or logically. There is no geological setting and the analytical techniques come after a sample description/petrography which mentions mineral chemistry. Because of this organization the reader gets a bit lost. A reorganization of the structure of the manuscript would assist the reader in following the flow of logic.

The introduction is short and to the point but could place the study in a better geological and overall geological process context. The geological background is limited in the introduction and there is no geological setting section. The authors only come back to this in the discussion. Rock types adjacent to the BIF and hosting it are not described at all in the manuscript and the geology of the region, area and even mine are not described. This significantly detracts from the work. This should be added even if briefly. 

The methodology followed is appropriate. Only one sample was investigated which may not be totally representative and so not totally sufficient. Are the authors sure that this is totally representative of the Changyi Metallogenic Belt? The data is of a good quality.

The language and phrasing are adequate for the most part, but there are grammatical errors. These are noted and largely corrected, or suggestions made regarding them in the annotated PDF. The data are of a good quality. The figures are good quality which show appropriate data. Some more info could be provided for some figures, such as mineral abbreviations. Incorrect figure numbers are referred to in the text in places, particularly for Fig. 3b (page 5). 

The cited references are appropriate to the work done. The interpretations and inferences are largely consistent with the data presented. The conclusions follow on from the interpretations made. What would be useful is to describe some shortcomings of the study and areas that require more detailed or further research. The conclusions are adequate to the findings and data.

Many of the strengths and weaknesses of the manuscript are mentioned above and addressing them can only help strengthen the manuscript. Overall the manuscript is of a fair quality and requires some major additions and corrections in places as highlighted above.

Specific comments are given in the annotated PDF version of the manuscript.

Author Response

Reviewer 1

Summary

This manuscript examines the anti-perthite and meso-perthitic textures in feldspar within BIF of the Zhengjiapo Mine in the Changyi Belt which forms part of the Jiao-Liao-Ji Belt (JLJB) of northeastern China. Using a sample of the BIF and determining peak and post-peak metamorphic assemblages, undertaking geochemistry of a homogenous leucocratic portion of the BIF, and doing mineral chemistry the P-T conditions the BIF were subjected to were determined. The study also makes use of two-feldspar geothermometry by visually determining the ratio of Na-feldspar to K-feldspar in the anti- and meso-perthitic feldspars and thereby determines the temperatures to which the BIF was subjected. These P-T conditions are used to highlight UHT conditions, higher than previous determinations of metamorphic conditions. This, along with age data not reported in this manuscript, but done as part of the study, help constrain the tectonothermal evolution of this portion, at least, of the JLJB.

The data used are appropriate to constrain the P-T conditions. The petrography, however, is poorly done and confusing and leaves the reader somewhat confused as to the mineral assemblages and what compositions belong with what. It’s also not clear how the authors determined the different mineral assemblages (M2’ and M3). Was it based on associations or textures? No geological setting is given and so it is difficult for readers to place the area or samples in a proper geological or tectonic context or even care why this methodology was done. This should be done more extensively upfront.

Reply: We add a description for the division of metamorphic stages in the section “3. Metamorphic stage and mineral chemistry” in revised MS, please see line 103.

 The findings add to the literature to constraining the P-T conditions of the JLJB. The manuscript is short and concise. In fact, too short and concise and leaves out some details that will help the reader place the study in a greater geological context or appreciate its significance. There are numerous examples of grammatical errors which detracts from the work and the science. Generally, the manuscript draws appropriate conclusions and interpretations from the data.

Reply, thanks for positive comments, we have added the introduction of geological background and division of the metamorphic stages to make it easier for readers to understand this study.

General comments

The subject matter is of general interest to Minerals readers, particularly those in East Asia, but does have a global relevance with regards to the metamorphism of BIF ores and using various techniques to determine the P-T conditions of metamorphism.

Reply, although not much, we present few examples worldwide related to BIF ore, essentially, there are only two types of mineral assemblage record UHT metamorphism, i.e coexisting exsolution textures of clinopyroxene and orthopyroxene in magnetite quartzites and the existence of antiperthite, perthite and mesoperthite. The temperature conditions are all calculated by using traditional thermometers up to now.

The interpretation adds important information to the tectonic development of the JLJB but the reader should be given the geological context better. The title is appropriate to the study and its content.

Reply, we have added the introduction of geological background in the chapter “2. Geological background and Sample description” of the revised MS.

The authors have gone for a short, concise manuscript but at the expense of giving information that would highlight the geological context and significance of the findings to not only the region but to other metamorphosed BIFs globally. A brief geological setting and overview is not given. Not all readers are familiar with the Jiao-Liao-Ji belt. The geology, overall setting, and tectonic, metamorphic and deformational history of the JLJB should be given so that the reader has an idea as to the geological context, when the JLJB formed, when the BIF was likely deposited and when metamorphic events occurred and what their tectonic or geodynamic setting have been postulated to have been. This would place the findings of this study in a better context and highlight its relevance to other metamorphosed BIFs around the globe. As such, currently as the manuscript is structured, the reader needs to gain this information piecemeal from reading other articles referenced throughout the manuscript. Many readers would not be bothered trying to chase down a number of other articles to gain an overview of the geological setting and relevance of the findings given in this study. It should, then, be briefly given in this manuscript up front. International readers may be frustrated by a lack of a proper geological context. As an example, groups are mentioned and given with no mention of their rock types, again. Short and concise is fine, but the reader needs to be given some information to make their reading worthwhile.

Reply, we have added an overview of geological background in the chapter “2. Geological background and Sample description” of the revised MS.

The pressures measured of 5.2-7.2 kbar are not consistent with collisional tectonics being too low. Collision would lead to thicker crust of 50 to 70 km thickness (~13 to 20 kbar)

Reply, the sample we studied only recorded this pressure under UHT condition, perhaps this is just the product of isothermal decompression, more research is needed in the future.

The petrographic or sample description could be better organized and more descriptive detailing mineral assemblages. The description should be split into the ore-rich domains first and then the felsic-rich (or leucocratic) domains. Microperthite and antiperthite (strictly speaking these are textures) are assigned to the M2 metamorphic stage, but no reasoning or justification for this is given. How is it known that these minerals form part of the M2 peak metamorphic assemblage? A description of the metamorphic assemblages should be given and descriptions and explanations as to how these were determined from the textures, mineral assemblages and associations. Different textures and assemblages which indicate when these different minerals formed as part of differing metamorphic stages should be described better and in more detail. It would be helpful to clearly state what the M2 and M3 mineral assemblages are at the end of the petrographic description section as this relates to what comes later in the pseudosection modelling.

Reply, thanks for detail comments, we added description in the chapter “3. Metamorphic stage and mineral chemistry”.

There was an error in reading off the pressures and temperatures determined from the pseudosection for M2’ conditions (Fig. 3b in the manuscript). This may influence which pressures were used to determine the temperatures using the two-feldspar geothermometer. As such, this may lead to some errors in the temperatures and pressures reported for the UHT metamorphism. The authors are requested to please check this.

Reply, the temperature calculated using two-feldspar geothermometer at stage M2 is independent of pressure. Since Opx is not found at M2 satge,which should associated with feldspar before the formation of antiperthite/mesoperthite exsolution texture, and the pressure at M2 stage cannot be constrained. The pressure of M2 stage, thus, is taken from that of M2’ stage and represented the lowest pressure of M2 stage. Above paragraph is added in the revised MS, please see lines 207-210.

The manuscript initially does not flow well or logically. There is no geological setting and the analytical techniques come after a sample description/petrography which mentions mineral chemistry. Because of this organization the reader gets a bit lost. A reorganization of the structure of the manuscript would assist the reader in following the flow of logic.

Reply, we have added an overview of geological background in the chapter “2. Geological background and Sample description” of the revised MS.

The introduction is short and to the point but could place the study in a better geological and overall geological process context. The geological background is limited in the introduction and there is no geological setting section. The authors only come back to this in the discussion. Rock types adjacent to the BIF and hosting it are not described at all in the manuscript and the geology of the region, area and even mine are not described. This significantly detracts from the work. This should be added even if briefly.

Reply, we have added a paragraph to the introduction to provide more detailed information on lithological association and rock types.

The methodology followed is appropriate. Only one sample was investigated which may not be totally representative and so not totally sufficient. Are the authors sure that this is totally representative of the Changyi Metallogenic Belt? The data is of a good quality.

Reply, at least it preserved UHT metamorphic conditions of the BIF iron ore experienced in the Changyi Metallogenic Belt although it might be local case. More studies will require for the entire mineralized zone.

The language and phrasing are adequate for the most part, but there are grammatical errors. These are noted and largely corrected, or suggestions made regarding them in the annotated PDF. The data are of a good quality. The figures are good quality which show appropriate data. Some more info could be provided for some figures, such as mineral abbreviations. Incorrect figure numbers are referred to in the text in places, particularly for Fig. 3b (page 5).

Reply, thanks for pointing out the problem. Grammatical errors are checked, and incorrect figure numbers are corrected. Mineral abbreviation is cited from [14]. Figure caption is checked and added information as needed, see figure 1 and 2 captions.

The cited references are appropriate to the work done. The interpretations and inferences are largely consistent with the data presented. The conclusions follow on from the interpretations made. What would be useful is to describe some shortcomings of the study and areas that require more detailed or further research. The conclusions are adequate to the findings and data.

Reply, thanks for comments. The conclusion part supplements the deficiency of this study and further research needs to be done.

Many of the strengths and weaknesses of the manuscript are mentioned above and addressing them can only help strengthen the manuscript. Overall the manuscript is of a fair quality and requires some major additions and corrections in places as highlighted above.

Reply, many thanks for all suggestions.

Specific comments are given in the annotated PDF version of the manuscript.

Reply, all comments have been carefully redacted.

Reviewer 2 Report

The manuscript submitted by Chen et al. illustrated the exsolution textures and equilibrium conditions of antiperthite and mesoperthite occur in felsic domains of the Zhengjiapo BIF, Changyi metallogenic belt, North China Craton, and concluded that the BIF is involved in the collisional tectonic process in Paleoproterozoic to have occurred in the Jiao-Liao-Ji orogenic belt. The manuscript reported that an ultrahigh temperature–medium pressure condition of Precambrian BIF and could be of scientific importance in the tectonic setting of BIFs. Overall, this manuscript present original data, and is quite interesting, well organized, well written and suitable to publish in Minerals. Nevertheless, there are still some issues that need the authors to explain and improve. This paper needs a minor revision before its acceptance. Specific comments are listed as follows. 

(1) Could you try to explain the geological significance in detail of the ultrahigh temperature–medium pressure condition of Zhengjiapo BIF in 5.1. Metamorphism of BIF iron ore?

(2) The abbreviations in Figure 1a and abbreviations of minerals in Figure 2 should be described in detail and the specific meanings should be written in the figure captions, such as TNCO, EB, WB, Qz and Ap.

(3) Please carefully check the area represented by metamorphic rock associated with BIF in Figure 5a.

Author Response

Reviewer 2

Comments and Suggestions for Authors

The manuscript submitted by Chen et al. illustrated the exsolution textures and equilibrium conditions of antiperthite and mesoperthite occur in felsic domains of the Zhengjiapo BIF, Changyi metallogenic belt, North China Craton, and concluded that the BIF is involved in the collisional tectonic process in Paleoproterozoic to have occurred in the Jiao-Liao-Ji orogenic belt. The manuscript reported that an ultrahigh temperature–medium pressure condition of Precambrian BIF and could be of scientific importance in the tectonic setting of BIFs. Overall, this manuscript present original data, and is quite interesting, well organized, well written and suitable to publish in Minerals. Nevertheless, there are still some issues that need the authors to explain and improve. This paper needs a minor revision before its acceptance. Specific comments are listed as follows.

Reply, thanks for positive comments.

(1) Could you try to explain the geological significance in detail of the ultrahigh temperature–medium pressure condition of Zhengjiapo BIF in 5.1. Metamorphism of BIF iron ore?

Reply, since no complete P-T path is constructed, it is difficult to explain geological significance in detail with this limited data.

(2) The abbreviations in Figure 1a and abbreviations of minerals in Figure 2 should be described in detail and the specific meanings should be written in the figure captions, such as TNCO, EB, WB, Qz and Ap.

Reply, mineral abbreviation is cited from [14], see Figure 1 caption. TNCO, EB, WB etc. are all added explanation in the figure caption as well as in the text.

(3) Please carefully check the area represented by metamorphic rock associated with BIF in Figure 5a.

Reply, have been down.
